# Genetic Engineering of *Agrobacterium* Increases Curdlan Production through Increased Expression of the *crdASC* Genes

**DOI:** 10.3390/microorganisms12010055

**Published:** 2023-12-28

**Authors:** Matthew McIntosh

**Affiliations:** Institute of Microbiology and Molecular Biology, IFZ, Justus-Liebig-Universität, 35292 Giessen, Germany; matthew.mcintosh@mikro.bio.uni-giessen.de

**Keywords:** curdlan, β-glucan, capsular polysaccharide, nitrogen limitation, genetic engineering, industrial microorganism, gene expression

## Abstract

Curdlan is a water-insoluble polymer that has structure and gelling properties that are useful in a wide variety of applications such as in medicine, cosmetics, packaging and the food and building industries. The capacity to produce curdlan has been detected in certain soil-dwelling bacteria of various phyla, although the role of curdlan in their survival remains unclear. One of the major limitations of the extensive use of curdlan in industry is the high cost of production during fermentation, partly because production involves specific nutritional requirements such as nitrogen limitation. Engineering of the industrially relevant curdlan-producing strain *Agrobacterium* sp. ATTC31749 is a promising approach that could decrease the cost of production. Here, during investigations on curdlan production, it was found that curdlan was deposited as a capsule. Curiously, only a part of the bacterial population produced a curdlan capsule. This heterogeneous distribution appeared to be due to the activity of Pcrd, the native promoter responsible for the expression of the *crdASC* biosynthetic gene cluster. To improve curdlan production, Pcrd was replaced by a promoter (PphaP) from another *Alphaproteobacterium*, *Rhodobacter sphaeroides*. Compared to Pcrd, PphaP was stronger and only mildly affected by nitrogen levels. Consequently, PphaP dramatically boosted *crdASC* gene expression and curdlan production. Importantly, the genetic modification overrode the strict nitrogen depletion regulation that presents a hindrance for maximal curdlan production and from nitrogen rich, complex media, demonstrating excellent commercial potential for achieving high yields using cheap substrates under relaxed fermentation conditions.

## 1. Introduction

Many bacteria produce extracellular polysaccharides (EPS) that are involved in attachment. EPS enable cell–cell attachment, facilitating the formation of flocs (rafts of aggregated cells in aqueous environments) and biofilms through enhancing attachment to surfaces in soils or on plants and animals. In pathogenesis they may confer resistance to host defenses [1], thereby promoting intracellular survival [2]. In non-pathogenic contexts, they may increase access to nutrients and protect against environmental stresses (e.g., desiccation and toxic agents) [3,4]. Some EPS are hydrophilic or gel forming polymers and are loosely adherent to the bacterial surface forming a slime layer (e.g., xanthan from *xanthomonas campestris* and succinoglycan from *rhizobia* and *agrobacteria* species). Others form a discrete capsule, which overlies the cell surface but is not attached to it (e.g., hyaluronan from *streptococcus* species and *pasteurella multocida* [5,6]). In *azotobacter vinelandii* a calcium alginate gel forms a coat around the resting cyst stage [7]. Some bacteria are known to produce several different EPSs simultaneously, such as *agrobacterium tumefaciens*, which can produce at least five [8].

A few EPS are water-insoluble, notably the (1-4)-β-glucan, cellulose, e.g., *gluconacetobacter xylinus* [9], *agrobacterium tumefaciens* [8], species of *pseudomonas*, *achromobacter*, *alcaligenes*, *rhizobium*, *azotobacter* [10], members of the enterobacteriaceae [11,12], and as a sheath around filamentous cyanobacteria [13].

Curdlan, a linear (1→3)-β-D-glucan, is another example of a water insoluble EPS. This polysaccharide is produced by various *agrobacterium* species [14], some rhizobium isolates [6], the gram-positive *cellulomonas flavigena* [15,16] and an isolate of *rahnella variigena*, a member of the enterobacteriaceae family [17]. Interestingly, in all cases cited above, curdlan production in the laboratory requires N-limited culture medium with ample carbohydrate, which fits with the general observation that the production of EPS [18,19,20,21,22], as well as the internally stored polyhydroxyalkanoates [23] are frequently induced by nutritional stress, e.g., N, P or S limitation. It is nonetheless remarkable that the capacity for curdlan production is not restricted to a single class or even a phylum (e.g., gram-negative), but rather to a handful of bacteria that seem related only by their soil habitat. So far, curdlan production has only been detected in rhizosphere-dwelling bacteria, and is in every case regulated by N-limitation.

The biological role of curdlan has not been demonstrated, but it has been suggested to help form a biofilm that protects against predation, starvation, and desiccation [8]. Such a role is consistent with microscopic observations of *C. flavigena* [24], a soil-dwelling, curdlan-producing bacterium that colonizes and decomposes plant material using cellulose as a source of carbohydrate. However, little information is available on the form of nascent curdlan, its deposition, and the ongoing nature of its interaction with the cell surface. Curdlan is occasionally referred to in the literature as a capsule, although clear evidence for a capsular form has not been provided. Curdlan-producing *agrobacterium* strains grown on glucose-rich yeast extract agar medium produce a coherent, aniline blue-staining, pellicle which may be stripped from areas of confluent bacterial growth [14], giving the appearance of curdlan deposition as a disorganized mass at the colony surface. However, such an appearance does not rule out the possibility of a capsular form. One possibility is that curdlan is produced as a discrete capsule by some cells located strictly in the periphery of a biofilm. This scenario takes into account the strong dependency of curdlan production on oxygen availability [25]. Peripheral cells would, in this way, have the role of providing curdlan as a protecting layer for the whole population. Alternatively, it is possible that upon production, curdlan helps to form a biofilm which engulfs the colony and forms a nebulous matrix of a biofilm without any discrete individual cell encapsulation. Only the former scenario would justify the formal designation of curdlan as a capsule according to the literature [26]. As one of the goals to this study, I wanted to investigate curdlan production by *agrobacterium* species, in particular to see whether curdlan was also deposited as a discrete capsule on this bacterium.

A single molecule of curdlan can be quite large, consisting of up to 12000 polymerized glucose residues [27]. Its size, as well as the unique, unbranched (1→3)-β-D-glucan structure confers upon curdlan its unique gelling and high water-retention properties. There have been many reviews of the large body of literature covering the chemistry and physico-chemistry of curdlan, its pharmacological and immunomodulation activities, and its commercial applications [21,27,28,29,30,31,32]. Curdlan is widely used, for example, as an additive to cement (super-workable concrete) for the construction industry, an additive to tablet formulations in the pharmaceutical industry, as well as an antiviral agent in immunotherapy and numerous applications in tissue engineering, drug delivery, and other biomedical applications. Moreover, curdlan is one of the most desirable edible polysaccharides because it lacks any unpleasant taste and because of its versatile applications as a thickener and healthy fiber in the food industry. Unfortunately, the industrial use of curdlan is limited by its high cost of production. One way to reduce the cost of production is to develop higher curdlan-yielding strains through genetic engineering. Genetic engineering of *agrobacterium* strains to increase curdlan production was another goal of this study.

The genetic basis of curdlan production has been investigated using *agrobacterium* sp. strain ATCC31749. Earlier studies found that the essential genes include a biosynthetic gene cluster, *crdASC*, and *crdR*, which codes for a transcription regulator [33,34]. CrdS shows some similarity to beta-glycoxyl transferases with repetitive action patterns [34]. The extremely hydrophobic CrdS protein is thought to form a multimeric (possibly an octamer) complex located in the inner membrane. This location is stabilized by seven transmembrane helices and a central large hydrophilic cytoplasmic region which contains the substrate-binding and catalytic motif.

The *crdASC* genes form a gene cluster (i.e., all transcribed in the same direction) within 5 kb of the linear chromosome of *agrobacterium*. The upstream of the first gene (*crdA*) in the *crd* cluster is a strong promoter, referred to as *crdP* [35] or Pcrd. It was not known whether Pcrd is the only promoter for the *crdASC* cluster. Transposon insertions into any of the genes *crdA*, *crdS*, or *crdC* result in the inability to produce curdlan. Loss of curdlan production by the insertion disruption of *crdA* is interesting because of the implication that either *crdA* is essential for curdlan production and/or because of the polar effect of disruption of transcription of the downstream gene *crdS*. The functions of the protein products of the genes *crdA* and *crdC* are not known.

The gene *pssAG*, coding for a phosphatidylserine synthase, also affects curdlan yield [33]. It is likely that specific phospholipids are important for the activity of the synthase complex in the inner membrane of this gram-negative bacterium [36]. In addition, curdlan production appears to be strictly governed by complex gene expression regulation [27,35,37]. Upon encountering the environmental triggers for curdlan production (e.g., low nitrogen and abundant sugars) the transcription of the *crdASC* genes increases by almost 100-fold, and this depended upon phosphate accumulation and the stringent response (ppGpp) [37]. This feature of curdlan is an exception to the general expectation that EPS production and biofilm formation in most bacteria are typically regulated by three primary mechanisms: quorum sensing, cyclic dinucleotides and small non-coding RNAs [38]. Considering that curdlan is a polymer of glucose that is destined for the cell exterior, it seems reasonable that the expression of genes that incorporates this high-energy molecule into an exopolysaccharide should be highly regulated. So far, the only regulatory mechanisms governing curdlan production that have been reported operate at the level of transcription control. This is in contrast to another exopolysaccharide produced by *agrobacterium*, succinoglycan, which does not appear to be regulated at the level of transcription [8].

Curdlan has been intensively studied to increase production and lower costs during fermentation [25,35,39,40,41,42,43,44,45,46,47,48,49]. Two of the major considerations for improving production and lowering the costs are the complex native regulation controlling production and the specific nutritional conditions required during fermentation. These two considerations are related, since the specific conditions required during fermentation drive up the cost of production. Generally, the native regulation serves to restrain curdlan production until certain specific conditions are met. These include low nitrogen, high carbon sources, generous phosphate levels, and high aeration [27]. Some strains of *agrobacterium*, e.g., the well-known plant pathogen *A. tumefaciens* C58 that is used in plant genetic manipulations, do not produce detectable levels of curdlan [8], even though they carry the necessary genes (*crdASC* and *crdR*). Other strains, such as the *agrobacterium* sp. strain ATCC31749, have ‘suffered’ from spontaneous mutations that appear to have partially relaxed the regulatory control over curdlan production, leading to high yields [6,50].

Various genetic modifications of the strain ATCC31749 have generally resulted in not more than 10–50% improvement [35,45,47,51]. For example, Kim et al. [51] reported a mutant strain that was able to produce 76 g/l curdlan in a stirred tank reactor after 120 h of cultivation, while the parent strain produced 64 g/l under comparable conditions. One possible explanation was that a greater production rate of curdlan is not possible, i.e., that the production rate is limited by cellular metabolism capacities, uptake of carbon source, or lifetime of the bacterial cell. In that case, no amount of genetic engineering could greatly improve the yield. However, another possibility (tested in this study) was that a constitutive transcription of the *crdASC* genes might yield higher levels of curdlan—an approach that has not been attempted. To state the hypothesis succinctly, if the native promoter Pcrd that controls the expression of the entire *crdASC* gene cluster were to be replaced with a stronger, unregulated promoter, this might lead to a higher curdlan yield, given that curdlan production is highly dependent upon transcription control. Related to the question of whether increased transcription of the *crdASC* genes would increase production is whether a stronger synthetic promoter would relax the requirement for specific conditions (e.g., high oxygen, low nitrogen) and support high curdlan production independently of the nutritional conditions.

Thus far, there have been no reports in the literature where the region of the linear chromosome of *Agrobacterium* carrying the *crdASC* genes has been modified with the goal of increasing their expression. The key to engineering a higher transcription rate for the *crdASC* gene cluster is the knowledge of suitable promoters capable of strong transcription in *agrobacterium*. During investigations on transcription regulation in the alphaproteobacteria, a strong promoter named PphaP (promoter of gene *phaP*) from the alphaproteobacterium *rhodobacter sphaeroides* was found, which is responsible for production of the phasin protein which binds to the polyhydroxyalkanoate granule [52,53]. I decided to test whether curdlan production could be improved using PphaP in the industrially used strain *agrobacterium* sp. ATTC31749 and the ‘wild type’ *A. tumefaciens* strain C58. DNA sequence alignments using blastn from NCBI on the *crdASC* gene clusters from each of the strains reveals 98.72% nucleotide identity. This analysis was based on 5.3 kb which includes the *crdASC* genes and the 300 bp promoter region, using C58 (in NCBI: *agrobacterium fabrum* str. C58) and 31749 (in NCBI: *agrobacterium* sp. CGMCC 11546). The key conclusion of this comparison is that both 31749 and C58 carry a *crdASC* gene cluster. (However, C58 does not produce quantifiable curdlan levels under standard laboratory conditions.) Therefore, I wanted to see if the insertion of PphaP into the chromosome of each *agrobacterium*, replacing the native *crdASC* promoter (Pcrd) and driving the expression of the *crdASC* gene cluster, could increase curdlan production in each strain. A similar outcome in both strains could potentially lend greater weight to the conclusions.

## 2. Materials and Methods

Bacterial strains and plasmids. The strains used in this study were the curdlan producing *agrobacterium* sp. strain ATCC 31749, whose genome has been sequenced [54] and *agrobacterium tumefaciens* C58 [55]. Mutants were generated using a chromosomally inserted suicide plasmid (pK18mob2) [56]. For transferring plasmids into the *agrobacteria*, the *E. coli* strain S17-1 was used for conjugation. Plasmid construction was performed using the restriction digest sites indicated in Figure 1.

Upon homologous recombination with the chromosome, the suicide plasmid inserts an additional copy of the homologous fragment carried on the plasmid. For this approach, I used different fragments from the *crdASC* gene cluster. Firstly, to learn about the expression of this cluster, a copy of the promoter region of *crdA* (upstream 300 bp) plus the first 51 bp of the coding region of *crdA* (351 bp total) was fused to the gene for the reporter protein mVenus. This translational fusion construct was inserted into the chromosome using the suicide vector pK18mob2 and homologous recombination (see Figure 1A, suicide plasmid 1, reporter plasmid). Upon insertion, the CrdA::mVenus fusion provided a readout of *crdA* expression. Importantly, the insertion created an additional copy of the *crd* promoter (Pcrd) without interrupting the native *Pcrd*-*crdASC* expression and curdlan production.

Another fragment of *crdA* was used in the second and third suicide variants (see Figure 2B,C, suicide plasmids 2 and 3). For suicide plasmid 3, this *crdA* fragment (489 bp) was preceded by a fragment from the bacterium *R. sphaeroides* encoding the *phaP* promoter (PphaP, 127 bp). Upon homologous recombination with the chromosome, this suicide plasmid variant inserts PphaP upstream of the *crdASC* gene cluster, meaning that expression of the cluster is now dependent upon PphaP. The DNA sequences of the suicide plasmids used in this study are included in a text file as in Appendix A.

For the measurement of the activity of Pcrd and PphaP using the reporter gene for mVenus, the low copy plasmid pPHU231 was used, in which either Pcrd or PphaP was fused to the gene for mVenus, as previously described [52]. In the case of Pcrd::mVenus, the fusion was a translational fusion, and was the identical fragment which was used for the suicide vector pK18mob-Pcrd::mVenus (see Figure 1A). This fragment included the ribosome-binding site of *crdA* plus the first 27 codons of *crdA* (including the translational start). In the case of PphaP::mVenus, the fusion included the native ribosome binding site of *phaP* plus the first 13 codons of *phaP* (including the translational start). A comparison of the two ribosome binding sites reveals some similarity: CAGGAGACAAGGCAatg for *phaP* and GAGGAGTTGATTTCatg for *crdA* (ribosome binding sites are underlined).

Media and culture conditions. *Agrobacterium* strains were routinely grown at 30 °C for a minimum of 3 d on aniline blue agar (ABA) [34]. For liquid cultures (shaken at 130 rpm), strains were grown in Luria broth (LB, 10% Oxoid bacto-tryptone, 5% Oxoid yeast extract, 10% NaCl) or *agrobacterium* defined broth (ADB). ADB is medium that is weakly buffered with a phosphate solution, so that initial growth of *agrobacterium* is around pH 7, which then subsequently drops to around pH 5.5 at the onset of curdlan production [34,50]. ADB consists of 4% (*w*/*v*) glucose (or 4% mannitol), 20 mM KNO_3_, 2.8 mM K2HPO_4_, 12.8 mM KH_2_SO_4_, 11.5 mM Na_2_SO_4_, 1.25 mM MgCl_2_, 0.09 mM FeCl_3_, 0.05 mM MnCl_2_, 1.0 mM citric acid, adjusted to pH 7.0 with NaOH before autoclaving. (Sterile glucose was added to the salts solution after autoclaving.)

MOPS-buffered medium was previously used for growth of *sinorhizobium meliloti* in a defined medium [58] and also supported curdlan production by *agrobacterium*. The MOPS medium consists of 50 mM MOPS (adjusted to pH 7 with KOH), 55 mM mannitol, 5 mM Na glutamate (or 20 mM for excess N), 250 µM CaCl_2_.2H_2_O, 37 µM FeCl_3_.6H_2_O, 48 µM H_3_BO_3_, 10 µM MnSO_4_.H_2_O, 1 µM NaMoO_4_.2H_2_O, 0.3 µM CoCl_2_.6H_2_O, 2 mM K_2_HPO_4_, and 4 µM biotin.

Microscopy. Fluorescence microscopy: An Olympus BX50 microscope with a xenon lamp and on a Zeiss Universal light microscope with ultraviolet light (365 nm from an OSRAM HBO-100 lamp) were used. Images were captured with an RT Monochrome SPOT camera (Diagnostic Instruments Inc., Foxboro, MA, USA). Complexes of curdlan with the (1→3)-β-glucan specific, Aniline Blue Fluorochrome (ABF) (Biosupplies Australia Pty. Ltd., Bundoora, Australia) were detected by mixing one drop of a 7-day culture from ADB with one drop of a 0.01% ABF solution on a glass slide, covering with a coverslip, which was then firmly pressed onto the slide. Fluorescence from the ABF-curdlan complex was observed with excitation at 330–385 nm and emission at 420+ nm. For vital staining: 10 mL of a 4d-ADB culture was incubated with 5 mM 5-cyano-2,3-ditolyl tetrazolium chloride (CTC) [59] for 1 h on a shaker (130 rpm) at room temperature. A drop of the culture was prepared as described above for fluorescence of ABF-curdlan complexes. Intracellular CTC fluorescence was observed with excitation at 530–550 nm and emission at 590+ nm using the same optics as described above.

Electron microscopy. Cells from ADB cultures were harvested and washed in water by centrifugation at forces not exceeding 500 g. Samples were mounted on formvar-coated, 300-mesh copper grids, and negatively stained with 2.5% uranyl acetate for observations using a Philips 420 transmission electron microscope (TEM) at 100 kV. Images were captured with a GATAN camera, with a 20× magnification factor. For cell-sections, colonies were harvested from ABA and fixed in a solution of 5% glutaraldehyde and 4% para-aldehyde in 0.2 M calcium cacodylate, 0. 1 M K_2_HP0_3_ (pH 7.0), for one day, washed in 0.1 M cacodylate, and post-fixed in 1% osmium tetroxide in 0. 1 M cacodylate. The cells were dehydrated in a graded series of ethanol (50–100%) each for 15 min, followed by 100% acetone, infiltrated in a 25% solution of Spurr resin (Electron Microscopy Science) in acetone for 1 h, followed by 1 h incubations in 50% and 75% resin. Specimens were finally embedded in 100% resin overnight, and the resin polymerized at 60 °C for 30 h. Specimen blocks were sectioned at 0.5 µm thickness, stained with lead citrate-uranyl acetate and viewed under a Philips 420 TEM at 100 kV.

Particle counting. ADB cultures grown for 2 to 7 d were diluted (1 in 10) with 50 mM Na_2_HP0_4_/NaH_2_P0_4_ (pH 7.0) before analyzing with a particle counter/sizer (CDA-500, Model: F-520p, Sysmex) with a 100 gm aperture detector. Particles were measured at two ranges: 10–40 μm and 40–100 μm.

96-well plate assay. Sterile, transparent, flat-bottom 96-well microplates (Greiner Bio-One) were prepared by adding 100 µL of 1% agar (liquid, 60 °C) containing aniline blue and nutrients from ADB or MOPS medium to each well. After the agar had cooled to room temperature, 20 µL of ADB or MOPS medium inoculated with *agrobacterium* was pipetted onto the agar, and the plate was covered with a transparent plastic lid (with condensation rings, high profile, clear, sterile, Greiner Bio-One) and incubated in a Tecan reader (Infinite 200Pro (M Nano+)) at 30 °C up to 72 h. The Tecan reader was programmed to measure the cultures growing in the well automatically at 3 h intervals.

Curdlan extraction. Curdlan was harvested from liquid cultures of *agrobacterium* as follows: The liquid culture (1 L) was subjected to centrifugation at 4000× *g*, 10 min. The supernatant was discarded and the pellet was resuspended in 1 L water. NaOH was added to the cell suspension a final concentration of 0.5 M, and stirred for 5 min. The cells were removed by centrifugation at 10,000× *g* for 10 min. The supernatant was removed to a clean container carefully without disturbing the cell pellet. HCl (32%) was added in slowly to the supernatant in a step-wise fashion, with good mixing, until pH 7. Then, the neutralized suspension was centrifuged at 10,000× *g* for 10 min. The supernatant carefully removed and discarded. The almost-transparent curdlan pellet was washed by resuspending in water and centrifugation at 10,000× *g*. This step was repeated 3 times, before the curdlan was placed in a petri-dish and incubated at 40–50 °C for 2–3 days. The dried curdlan was then weighed.

## 3. Results

Microscopical analysis of curdlan producing *Agrobacterium* cells reveals curdlan capsule. The strain ATCC31749 routinely produces curdlan when grown in liquid minimal salts with limiting nitrogen for longer than 3 days, whereas a knock-out (KO) strain of *crdA* produced no detectable curdlan (<0.01 g/L). This 3-day delay in curdlan production fits with the knowledge that curdlan producing *agrobacterium* cultures undergo a two stage development: an initial stage involving cell growth, and a later stage capable of producing curdlan provided that nitrogen is limiting and a suitable carbon source is present [27]. The 3-day growth period was also necessary for the appearance of flocs of variable size which were visible to the naked eye in cultures of the curdlan producing strain, but not in those of the *crdA* mutant. From this observation, it is reasonable to suspect that curdlan production was associated with floc formation.

From a 7-day culture, the cells were examined using phase contrast microscopy. Two prominent morphological forms of cells were observable in the curdlan producing cultures. These were a thicker rod form averaging 1.5 × 3.0 μm, located in flocs (cell aggregates) and a slender rod form, averaging 0.5 × 2.0 μm that was planktonic (Figure 2A). Only the slender rod form was observed in cultures of the *crdA* mutant. When stained with the (1→3)-β-glucan-specific aniline blue fluorochrome (ABF) [60,61] only the larger form exhibited strong yellow/green fluorescence in a defined zone around the cell periphery (Figure 2B). These observations suggest that curdlan is deposited as a capsule and that production of the capsule is restricted to cells located in flocs. Since curdlan production commences primarily in the post-growth stage of fermentation, it is likely that floc formation is mostly through the recruitment of capsulated cells, rather than through growth of new cells within a floc.

Electron microscopy of curdlan capsules. The physical form of native curdlan on the *agrobacterium* cell surface was investigated by TEM of uranyl acetate stained preparations. At low magnification (9600×), flocs from the curdlan producing culture were heavily stained, as might be expected for encapsulated cells. Occasional cells appeared to contain incomplete capsules (Figure 2C). In contrast, the aggregates of cells from the *crdS* mutant culture were quite difficult to find (being very rare) and not heavily stained (Figure 2D). The fibrillary nature of the capsular material is evident in sections of embedded curdlan producing cells stained with lead citrate-uranyl acetate (Figure 2E) when compared with the naked surface of the *crdS* mutant cells (Figure 2F). The use of electron microscopy (Figure 2C–F) confirmed the deposition of curdlan as a capsule—defined as a polysaccharide layer that lies outside the cell envelope. In contrast, bacteria incapable of producing curdlan had a ‘clean’ surface. These analyses confirmed curdlan as a discreet capsule found only on the surface of curdlan producing cells.

Formation and organization of flocs. As noted above, the curdlan producing form was rarely planktonic. Rather, the curdlan capsule was usually confined to cell aggregates or flocs. This observation lead to the hypothesis that abundant flocs ought to appear in curdlan producing cultures, but not in cultures incapable of curdlan production. Floc formation was assessed using a particle counter capable of detecting flocs as particles. (The particle counter detects flocs as particles using a cut-off of 10 and 40 µm, meaning that all flocs outside of this range, both larger and smaller, are excluded). When grown in liquid minimal salts for 3 d on a rotary shaker, ATCC31749 contained a proportion of particles larger than 20 µm in diameter, while the *crdS* disruptant contained mostly particles with a diameter less than 20 µm (Appendix A). At 6 days, the particles in the ATCC31749 culture were mostly larger than 20 µm in diameter. However, the particles from the *crdS* disruptant did not increase in diameter. This result supported the hypothesis that curdlan production was indeed associated with floc formation, and that the loss of curdlan production is associated with failure to form high numbers of flocs in liquid cultures.

96-well plate assay reveals onset of curdlan production. How well does expression of the *crdASC* genes correlate with curdlan production? A tight correlation would provide more support for the idea that curdlan production is controlled by *agrobacterium* at the level of *crdASC* gene expression. On the other hand, a poor correlation would indicate that curdlan production was controlled by additional unknown factors. A previous study using transcription analysis found a 100-fold upregulation of the *crdASC* genes upon the transition to nitrogen starvation [37], which lends weight to a tight correlation. To recapture this result, as well as to have a more sensitive, rapid and cheaper technique for the detection of both gene expression and curdlan production, a novel assay capable of a high throughput was developed. Until now, detection of curdlan production had been performed qualitatively via growth on agar containing aniline blue (where curdlan producing colonies turn intense blue while non-producing colonies remain white) or quantitatively following the extraction of curdlan via alkaline solution, neutralization, washing, drying and weighing. The extraction method is relatively laborious and impractical for analyzing a high number of samples, while growth on standard aniline blue agar plates is impractical for obtaining temporal data. For developing this assay, I took advantage of the knowledge that curdlan accumulation in bacterial cultures is easily detectable using the (1→3)-β-glucan-specific dye aniline blue [27]. For the setup, 100 µL of 1% agar containing aniline blue and nutrients from ADB (*Agrobacterium* define broth) was placed in each well in a 96-well plate. The rational for using agar in the 96-well plate was to minimize the volume of liquid cultures. This reduces the problem of measuring growth in liquid cultures with strong floc formation and the associated variation between measurements. Once the agar was set, 20 µL of ADB inoculated with *agrobacterium* was pipetted onto the agar, and the plate was covered with a transparent plastic lid to prevent drying of the agar and incubated in a Tecan reader at 30 °C up to 72 h. The Tecan reader was set to measure the cultures growing on the surface of the agar in the well automatically at 3 h intervals.

For the 96-well plate assay, reporter strains were prepared (using C58 and ATCC31749) in which the gene for mVenus was fused (translational fusion) to *crdA* (see Figure 1 for details of the genetic construct). Using these strains, I specifically looked for correlation between the onset of *crdA* gene expression (detected as mVenus fluorescence) and the onset of curdlan production (as detected via aniline blue/curdlan fluorescence).

Absorbance at 600 nm (providing an indication of cell density, Figure 3A) was strongly impacted by curdlan accumulation in strain ATCC31749 at around 30 h, since binding between aniline blue and curdlan results in an intense blue colour that also absorbs light at 600 nm (Figure 3A). The large error bars are likely because of the irregular colony surface that accompanies curdlan production. Growth of C58 was weaker than that of ATCC31749.

Curdlan binds to aniline blue, and the complex emits red fluorescence at 548 nm when excited by light at 515 nm. Fluorescence from the aniline blue-curdlan complex was apparent in cultures of ATCC31749 after 30 h, but not in cultures of C58 (Figure 3B). Interestingly, fluorescence from mVenus (Figure 3C) was apparent in cultures of both ATCC31749 and C58 around 30 h, although a weaker basal level of mVenus fluorescence was apparent slightly earlier, at 24 h for ATCC31749.

Since C58 did not produce curdlan under these conditions, the data suggests that the expression of the genes *crdASC* was too low to support curdlan production. Alternatively, the critical step controlling the onset of curdlan production lies elsewhere, so that the expression of these genes is not the deciding factor for curdlan production in this strain. This was not the case for ATCC31749, where the onset of strong *crdASC* gene expression (at about 30 h) correlated relatively well with the onset of curdlan accumulation (as would be expected if curdlan production was controlled by *crdASC* expression). Thus, curdlan accumulation correlated with *crdA* gene expression in ATCC31749, but not in C58. Altogether, the newly developed assay appeared to be reliable for sensitively detecting curdlan accumulation and gene expression at a high throughput level.

The *R. sphaeroides* promoter PphaP is much stronger than Pcrd in *Agrobacterium*. Since curdlan production correlates with *crdASC* gene expression, one possibility for enhancing curdlan production is the enhanced expression of the *crdASC* genes. For this goal, the replacement of the native *crdASC* promoter (Pcrd) with a constitutively active promoter could suffice. I came across an obvious candidate promoter during investigations on transcription regulation in the Alphaproteobacteria. This candidate was PphaP from *R. sphaeroides* [52], which was very active in *A. tumefaciens* C58. To compare the activity of the two promoters, PphaP and Pcrd, in *agrobacterium*, the reporter gene for mVenus was fused to each of the promoters on the low copy plasmid pPHU231 (see materials and methods for details). The *agrobacterium* strains C58 and 31749 carrying the KO plasmid were used, since this avoided curdlan production and the resulting floc formation, which tends to interfere with fluorescence measurements and increase standard deviation. Furthermore, a variety of media was used. PY medium contains peptone and yeast extract (a rich source of nitrogen), while the MOPS medium contains a defined, growth limiting level of nitrogen. Apple extract (10%) represents a cheap and locally available source of nutrients. For details see materials and methods. In C58, the activity of Pcrd was low during growth in PY (Figure 4A) and also in MOPS medium. In contrast, the PphaP promoter activity was much stronger in all media (Figure 4B). A similar pattern was observed with the Pcrd and PphaP promoters in 31749 (Figure 4C,D). It is unknown why the presence of glucose or apple extract stimulated PphaP activity in PY medium. Also unexpected was the observation that Pcrd was rather weak in strain C58. This is probably related to the setup conditions, where 100 µL of liquid medium was used. The previous setup used in Figure 3 involved only 20 µL of liquid culture overlaying 100 µL of agar, likely providing better access of the bacterial culture to oxygen, which is known to be important for curdlan production [27], and is likely more critical for strain C58 than for the industrial strain 31749. Also in 31749, the Pcrd promoter was virtually inactive in PY medium, while in the MOPS medium, the Pcrd promoter was not active at 21 h, but did show some activity at the 45 and 72 h time points (Figure 4C). This fits well with the data in Figure 3C (the main difference being that the *crdA*::mVenus fusion was carried on the chromosome in Figure 3C, and on a low copy plasmid in Figure 4). This also fits with the knowledge that the expression of the *crdASC* genes is strongly activated upon N depletion [37]. In summary, these data demonstrate that PphaP was far more active than Pcrd in all the growth media tested.

PphaP activates curdlan production. At this point, I was interested in seeing how the PphaP promoter affected curdlan accumulation. Specifically, would the use of PphaP to drive the expression of *crdASC* lead to a higher production of curdlan? To test this the 96-well plate assay was used.

The data shows that the C58 WT (Figure 5A) did not produce curdlan (as was also shown in Figure 3B), while the insertion of PphaP resulted in curdlan production. Also in 31749 the presence of PphaP clearly enhanced curdlan production so that curdlan accumulation was apparent much earlier, at around 10 h of incubation, while in the 31749 WT, the onset of curdlan accumulation was apparent after 30 h of incubation (Figure 5B). Fluorescence reached maximum levels at around 400 fluorescence units in 31749, possibly because of exhaustion of the carbon source (mannitol).

Curdlan production requires both the expression of the *crdASC* genes and a suitable carbon source. Does the insertion of PphaP mean that curdlan production is independent upon N availability? When grown on PY agar with aniline blue but without any additional carbon source, the colonies did not stain blue, indicating the absence of curdlan production (Figure 6A). However, when glucose was added (1% final concentration), both the strains carrying PphaP, but not the WT strains, stained blue during growth on agar, indicating curdlan production (Figure 6A). This result implies that curdlan production requires at least two criteria: high expression of the *crdASC* genes as well as the presence of a suitable carbon source. Only one of these two conditions does not lead to curdlan production. Importantly, when Pcrd was replaced with PphaP, the bacteria appeared capable of curdlan production regardless of whether nitrogen was limiting, provided that a suitable source of carbon was available.

Apple extract is a suitable carbon source for curdlan production. Given that one of the major costs associated with curdlan production is the carbon source, I looked for realistic alternatives to the typical carbon sources. Apple juice (100%) was obtained from a local apple juice factory, sterilized via autoclaving and added to PY agar in increasing concentrations. At 1% (final concentration), a low level of curdlan production was apparent judging by the aniline blue stain (Figure 6A). Production was improved with 3%, 10% and 30% apple extract, with 10% supporting the most intense blue color formation.

To quantify curdlan, the bacteria were grown in liquid cultures for 3 days with ADB using glucose as a carbon source. The results show that curdlan production by C58 was not detected after growth in ADB, while C58 PphaP produced about 0.1–0.2 g/L curdlan. Curdlan production by 31749 in ADB was also about 0.2 g/L, while production by 31749 PphaP in ADB reached 1.6 g/L. Thus, under these conditions, PphaP improved curdlan production about 8-fold in strain 31749 and forced C58 to produce about 0.2 g/L curdlan under conditions where it would normally not produce any.

## 4. Discussion

The work here shows, firstly, that curdlan is produced by *agrobacterium* strains and deposited as a capsule that is responsible for strong cell–cell aggregation and floc formation in liquid cultures. Secondly, the work reveals a genetic engineering approach that enables approximately 8-fold higher curdlan yields from the commercially useful strain *agrobacterium* sp. 31749, and forces the wild type strain *A. tumefaciens* C58 to produce significant levels of curdlan where it previously produced none.

Curdlan-producing *agrobacterium* colonies grown on agar medium with a sufficient carbon source and aniline blue produce curdlan as a stable, intensely blue-staining layer which may be stripped, along with the cells, from the colony surface [14]. This suggests close proximity between cells and curdlan, and that curdlan is somehow adhesive and involved in attachment between cells. However, the relationship between curdlan and the bacterial cell surface had not been clearly defined. Scanning electron microscopy of the Gram positive curdlan producing bacterium *cellulomonas* indicated that curdlan may be deposited as a capsule [16]. However, staining with aniline blue could not support a capsular formation by *cellulomonas* since the native form of curdlan produced by this bacterium does not stain with aniline blue. It must first undergo alkaline extraction before staining, indicating a fundamentally different native form to curdlan produced by *agrobacterium* [15]. In this study, it was shown that *agrobacterium* deposits curdlan as a capsule, using the (1→3)-β-glucan-specific aniline blue fluorochrome (ABF). Additionally, curdlan was negatively-stained by uranyl acetate and thereby visible using TEM. Both ABF-induced fluorescence images and TEM showed that the curdlan capsule is tightly associated with the cell surface. Neither of these features was observed with a curdlan deficient mutant (*crdS*-) or in cultures of strains grown in medium that does not elicit curdlan production, i.e., Luria broth.

Cell aggregates depend upon curdlan production, and these aggregates appear to be via interactions between curdlan-encapsulated cells. The data is consistent with a scenario where cells produce a capsule, whereupon these encapsulated cells are recruited to the aggregate on the basis of curdlan–curdlan interactions. In other words, after producing curdlan, the cell randomly encounters a neighboring cell that has also produced curdlan. The two become attached and then encounter additional cells that have produced curdlan, and in this way the aggregates grow larger through cell recruitment over time (Appendix A). Studies on the soil-dwelling gram-positive *C. flavigena* also found a correlation between aggregation and curdlan production. Furthermore, on subculturing *C. flavigena* on a nitrogen-rich medium lacking a carbohydrate source, disaggregation occurs and the curdlan disappears, suggesting that the curdlan is mobilized as a carbohydrate source [15,16]. Consumption of curdlan by *agrobacterium* has not been observed.

*A. tumefaciens* is considered one of the more important, well-studied plant pathogens, being a tumorigenic pathogen responsible for crown gall disease with a wide host range [62]. So far, there is no indication that curdlan is necessary for pathogenesis. *Agrobacteria* strains are endemic in soils. Many are not necessarily pathogenic [63], and may exhibit long term survival even in the absence of host colonization [64]. Probably the ability to produce curdlan is related to long term survival, although this has never been demonstrated. Indeed, the *A. tumefaciens* C58 strain used in this study did not produce curdlan under standard laboratory conditions. Interestingly, when C58 was grown on apple extract, some curdlan production was apparent. Importantly, when C58 carried PphaP, strong curdlan accumulation was apparent (Figure 6A). It is likely that curdlan production is more highly regulated in C58 than in 31749 and that, in addition to nitrogen limitation, other environmental conditions (such as desiccation, or plant-derived molecules) are needed for the activation of curdlan production in C58.

The curdlan dependent aggregation phenomenon described here appear to enable the formation of microcolonies or flocs and biofilms in these nutritionally stressful situations. The curdlan capsule formed under these conditions may play a role in protection from desiccation by virtue of the impressive water-holding capacity of curdlan capsule, i.e., the encapsulated cell may act as a ‘spore’, as suggested in the case of cellulose-producing bacteria [11]. Moreover, curdlan could protect not only the producing cells, but also the entire community of cells from desiccation as well as soil predators such as phagocytic protozoans [65] and parasitic bdellovibrio species [66].

Interestingly, not all cells in the curdlan producing cultures were encapsulated. This heterogeneity within the population is somewhat similar for the production of another exopolysaccharide, galactoglucan, by another member of the family rhizobiacea, *sinorhizobium meliloti* [67]. The ability of clonal populations to display multiple phenotypes has been noted [68], and provides a number of advantages for survival such as bet-hedging and division of labor. Without knowing the role of curdlan in survival, it is difficult to say which of these options applies to curdlan. For example, curdlan forms a capsule, which may provide an advantage for the producing cell (e.g., protection against predation or dehydration). This could be a bet-hedging strategy, which can be viewed as a preadaptation of a subpopulation to a danger before it is encountered. However, curdlan also promotes cell–cell aggregation and formation of flocs, and could therefore be important for the formation of a biofilm (which benefits the whole population). In such a scenario, curdlan production by a subpopulation could be a form of division of labor. Both scenarios are not mutually exclusive, and more experiments are needed to clarify the role (if any) of heterogeneity in curdlan producing cultures.

Relevant to the goal of maximal curdlan production, such heterogeneity likely has the undesired effect of decreasing curdlan production, since not all individuals contribute. A more desirable outcome would be that all cells contribute to curdlan production. The use here of a strong, constitutively active promoter such as PphaP to drive the *crdASC* gene expression may well have contributed to the increase in curdlan production by removing the heterogeneic pattern of production, i.e., by forcing all cells to contribute.

The *phaP* promoter (PphaP) from *rhodobacter sphaeroides* controls the expression of the phasin gene *phaP*. Previous work on this gene found that it was strongly active in the exponential growth phase of *R. sphaeroides* [52]. Interestingly, upon heat shock, transcription of the *phaP* gene increased by over 20-fold [53], making the promoter of *phaP* one of the most active promoters that I have ever observed. Indeed, PphaP showed very high activity in *Agrobacterium* strains (Figure 4). In comparison, the native *crd* promoter (Pcrd) was not only much weaker than PphaP, it was also highly dependent upon the growth medium (e.g., nitrogen content). For example, the PY medium (containing peptone and yeast extract) did not support Pcrd activity, while PphaP was relatively strong. This made PphaP the ideal candidate for strong, unregulated expression of the *crdASC* genes. The outcome of this approach was a clear improvement in curdlan production (Figure 5 and Figure 6). However, curdlan production by strains carrying PphaP remained dependent upon the presence of a suitable carbon source (Figure 6A). This outcome implies that curdlan production is not only dependent upon expression of the *crdASC* genes, but also upon the presence of a suitable carbon source, which likely determines the metabolic status of the cell.

It is important that the KO strains did not produce curdlan (Figure 5 and Figure 6A). The significance here is that the suicide plasmid 2 (see Figure 1B) does not disrupt the *crdASC* genes upon insertion into the *agrobacterium* chromosome. Rather, the insertion causes transcription from Pcrd to be terminated by a transcription terminator, and thus the *crdASC* genes in the KO strains are not disrupted but rather transcriptionally inactive (i.e., isolated from Pcrd via the terminator). The outcome is that the KO strains do not produce curdlan, even in the presence of a suitable carbon source (Figure 6A). This fits with the idea that the *crdASC* gene cluster is indeed an operon with Pcrd as the promoter of the operon. Moreover, curdlan production requires both criteria: *crdASC* expression and a suitable carbon source. Nitrogen limitation is not necessary, so long as these two requirements are satisfied. This scenario is therefore ideal for the engineering approach used in this study. Despite genetically fixing the high *crdASC* gene expression, curdlan production will only follow once the bacterium encounters an adequate carbon source. This is useful for preparing starter cultures that are not encumbered with excessive curdlan production and cell aggregation.

How stable are the strains modified with a suicide vector? While such experiments are possible, I have not generated quantitative data to answer this question. However, Figure 6A provides qualitative data. In this study, the insertion of pK18mob2 into the chromosome was used for genetic modification following homologous recombination. pK18mob2 contains a kanamycin selection marker, meaning that the loss of this plasmid would mean the loss of kanamycin resistance. Furthermore, such reverting mutants would accompany a phenotype. For example, the KO strains lost their ability to produce curdlan and therefore appeared as white colonies on agar with aniline blue. When grown on agar lacking the antibiotic kanamycin, any appearance of blue colonies would indicate the spontaneous loss of this suicide plasmid as cells recovered the ability to produce curdlan. Thus, the presence of blue colonies mixed in with the white colonies would provide a qualitative measurement of the stability of the suicide vectors. That no blue colonies appeared among the KO strains is an indication that the suicide plasmid is highly stable. It is also worth noting that if, for some reason, the presence of the suicide vector in the chromosome of the curdlan producing strain is undesirable, its removal is possible (while maintaining the presence of PphaP) through a well-established method that uses the *sacB* gene [56].

## 5. Conclusions and Outlook

The work here describes a genetic engineering approach that enables higher curdlan yields. However, there is much work still needed to establish optimal conditions. Now that the high expression of the *crdASC* genes is fixed, which sources of N and C are most suitable for high curdlan yields? What concentrations of P and dissolved oxygen are optimal, and at what pH? Such questions have been intensively addressed for *agrobacterium* sp. 31749 [21,27,48], but these should be re-addressed for the genetically modified strains. I anticipate that the 96-well plate assay first described in this study could be a useful tool for answering these questions. Furthermore, which strains are most useful for high quality, high purity curdlan production? My approach using PphaP showed that curdlan production can be enhanced in both 31749 and C58 strains, despite the 1% genetic differences between the strains. Note that this difference likely accounts for different growth behavior (31749 reaches a much higher cell density in liquid cultures than C58, see Figure 3A) which could also account for the higher yield per liter of culture by strain 31749. However, strain C58 might be better at growth on various fermentation substrates, e.g., plant biomass.

Various plant biomasses have been tested as sources of carbon for curdlan production, such as sugar cane, cassava starch, wheat bran, orange peels and prairie cordgrass [41,45,49,69,70,71]. To my knowledge, however, apple extract has not been tested for curdlan production. Apples made up about 80–90% of all tree fruits harvested in Germany (https://www.destatis.de/EN/Themes/Economic-Sectors-Enterprises/Agriculture-Forestry-Fisheries/Fruit-Vegetables-Horticulture/_node.html (accessed on 7 August 2023)) and is therefore a local, cheap carbon source for curdlan production. In particular, the waste associated with apple storage, transport and pulp from juicing might be used for curdlan production. I found that apple extract added to agar at 10% supported what appeared to be a high yield of curdlan. It is even possible that apple extract might be superior to glucose as a carbon source, although this needs further research. Thus, apples could be cheap source of carbon for curdlan production. Combined with the new strains developed in this study that are not sensitive to nitrogen levels, new fermentation processes based on a variety of cheap plant biomass sources could dramatically decrease the cost of curdlan production, extending the use of curdlan in industry.

## 6. Patent

The Justus-Liebig Universität has filed a patent application (EP23210906.6, 20 November 2023) covering the modification of the *Agrobacterium* genome to increase expression of the *crdASC* genes to enhance curdlan production.

## Figures and Tables

**Figure 1 microorganisms-12-00055-f001:**
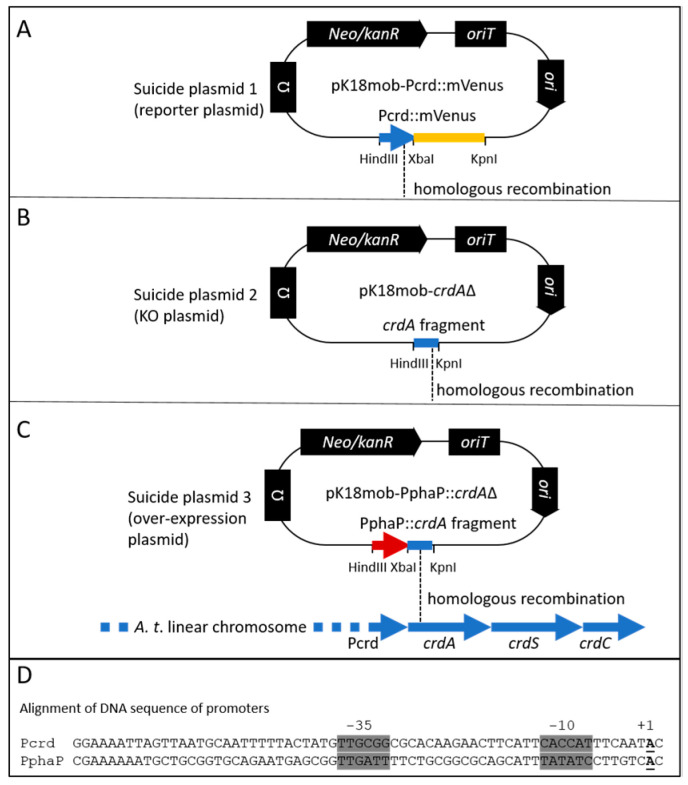
Genetic modifications of the *crdASC* locus. Three versions of the suicide plasmid (**A**–**C**) with highlighted differences for ensuring genetic modification at the *crdASC* locus. (**A**), the *crdA*::mVenus reporter construct (suicide plasmid 1, translational fusion) is inserted into the chromosome via homologous recombination (indicated with dashed line) using the PcrdA fragment fused to the gene for mVenus (in yellow). The insertion of this construct provides a read-out of *crdA* expression without disrupting the *crdASC* gene cluster. (**B**), suicide plasmid 2 includes a 507 bp fragment from the *crdA* open reading frame for guiding homologous recombination. The fragment includes the ribosome binding site and the translation start of *crdA* (total length of *crdA* = 1458 bp). Insertion of this construct effectively knocks out (KO) transcription of the *crdASC* gene cluster, since this cluster lacks now lacks a promoter, while the Pcrd controls only a fragment of *crdA*. (**C**), the strong PphaP promoter sequence is fused to the fragment of *crdA*. Insertion of suicide plasmid 3 via homologous recombination ensures overexpression of the *crdASC* gene cluster due to the insertion of the strong PphaP promoter directly upstream of *crdA*. (**D**), alignment of the DNA sequences of the *crd* and *phaP* promoters. Highlighted in grey are the −35 and −10 promoter elements. The transcription start of PphaP (+1, bold, underlined) was identified based on RNA sequencing data from *R. sphaeroides* [57]. The transcription start of Pcrd was based on Yu et al. [35].

**Figure 2 microorganisms-12-00055-f002:**
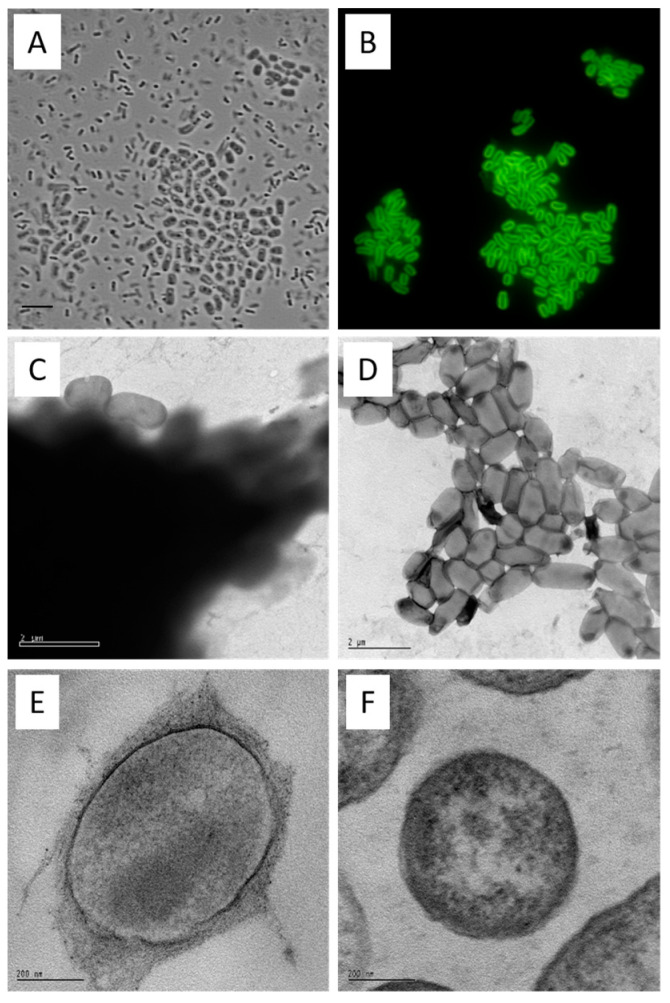
Curdlan forms a capsule. *Agrobacterium* sp. 31749 cells were grown for 7 days in *agrobacterium* defined broth (ADB) and subjected to microscopic analyses. Cells were stained with aniline blue fluorophore and photographed with bright field ((**A**), bar = 5 µm) and fluorescence ((**B**), excitation at 330–385 nm, emission at 420–475 nm). (**A**,**B**) photographs are from the same field. (Note that this sample was prepared by firmly pressing the glass slide and the object cover together, resulting in the disruption of some cells from the cell aggregates). Cells grown in ADB from strain 31749 (**C**) and 31749 *crdS* disruptant (**D**) were stained with uranyl acetate and viewed using electron microscopy (bar = 2 µm). These strains were also subjected to scanning electron microscopy ((**E**,**F**), bar represents 200 nm). The curdlan capsule appears as a dark cloud in (**C**), and an irregular outer layer in (**E**). The capsule is absent in a *crdS* mutant (**D**,**F**).

**Figure 3 microorganisms-12-00055-f003:**
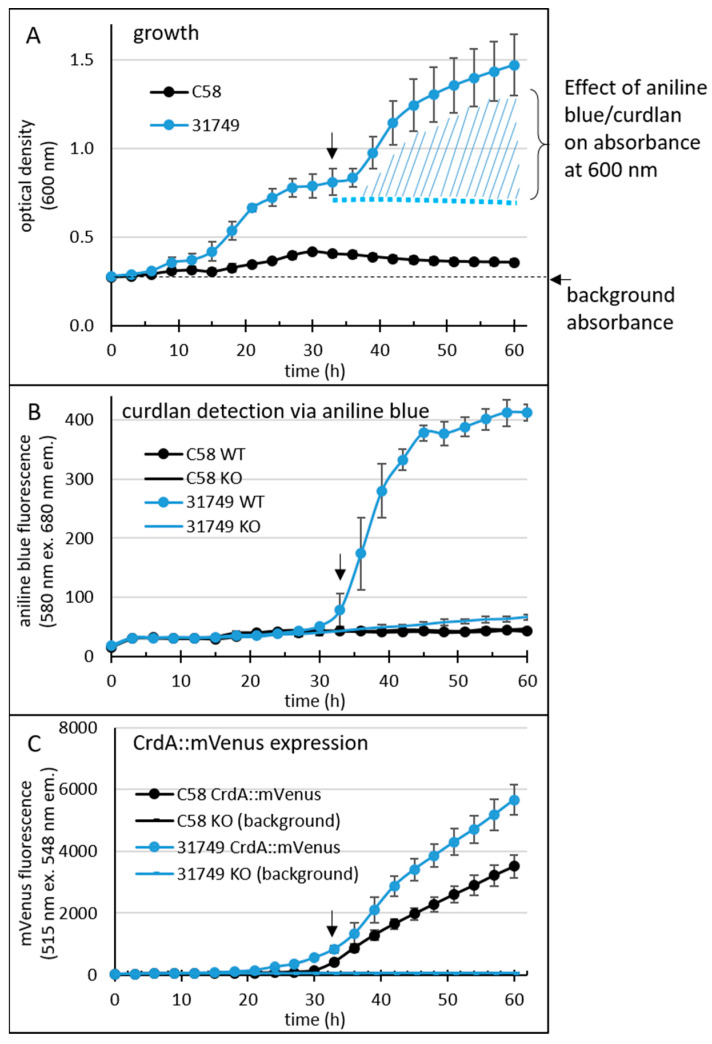
96-well plate assay reveals a temporal profile of the onset of curdlan production. *Agrobacterium tumefaciens* C58 (C58 WT) and *agrobacterium* sp. 31749 (31758 WT) were grown in a 96-well plate (8 wells for each strain). For the medium and other details of this setup, see the materials and methods. For the detection of curdlan, the medium contained 0.1 mg/mL aniline blue (0.01%). Additionally, a translational fusion between *crdA* and the gene for mVenus was inserted into the chromosome, allowing detection of *crdA* expression. Growth (**A**), fluorescence from aniline blue/curdlan (**B**) and fluorescence from mVenus (**C**) of the cultures was measured once every three hours for 60 h using an automated Tecan reader setup. Black arrows at approximately 33 h indicate the onset of curdlan production.

**Figure 4 microorganisms-12-00055-f004:**
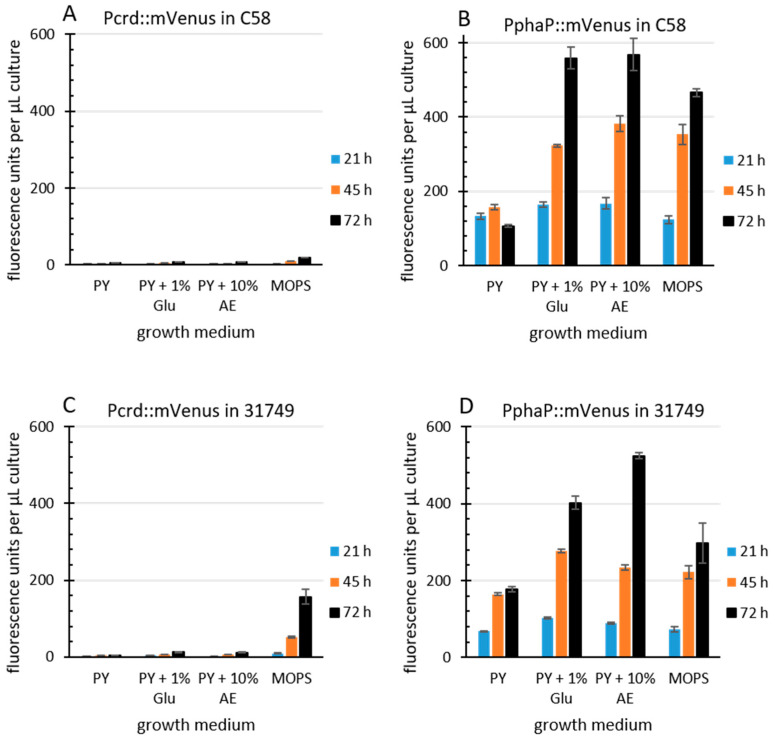
PphaP is more active than Pcrd. *Agrobacterium* strains ((**A**,**B**), C58, (**C**,**D**) 31749) incapable of curdlan production (via the pK-KO suicide vector) carrying the low copy plasmid pPHU231 containing the Pcrd::mVenus fusion (**A**,**C**) or the PphaP::mVenus fusion (**B**,**D**) were grown in a 96-well plate in various media. PY, peptone yeast, Glu, glucose, AE, apple extract (10%), MOPS, buffered minimal defined medium. Cultures were measured for mVenus fluorescence at the times indicated, and fluorescence per µL of bacterial culture is presented. Error bars indicate standard deviation from 8 well cultures.

**Figure 5 microorganisms-12-00055-f005:**
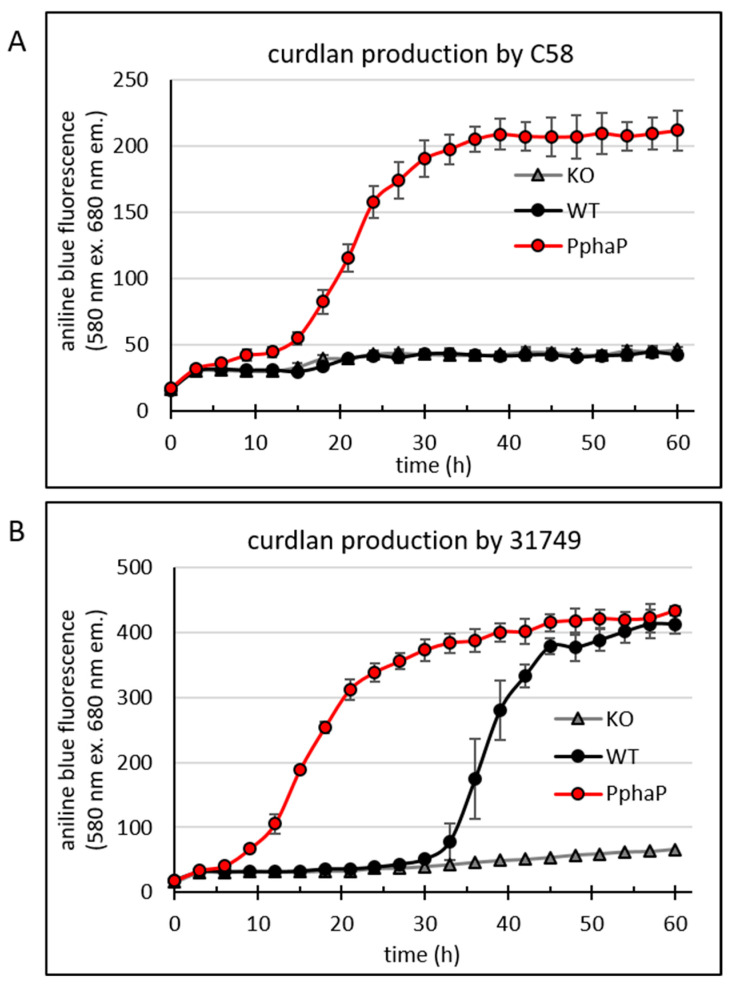
PphaP increases curdlan production in minimal medium. Bacteria were grown in a 96-well plate assay to reveal a temporal profile of the onset of curdlan production. Fluorescence from aniline blue/curdlan of the cultures was measured once every three hours for 60 h using an automated Tecan reader setup. For the details of this setup, see the materials and methods. Cultures include the C58-derivatives *crdA* knockout (KO), C58 wild type (WT) and C58 PphaP for (**A**) and likewise for the 31749-derivatives (**B**). Error bars indicate standard deviation from 8 wells.

**Figure 6 microorganisms-12-00055-f006:**
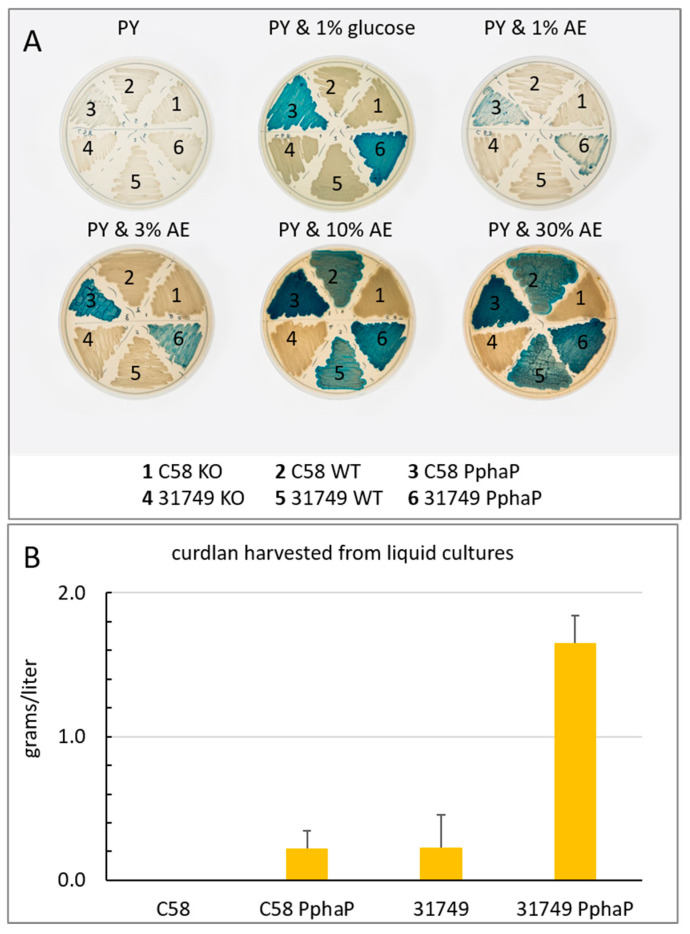
PphaP increases curdlan production from apple extract (AE) and in defined medium with glucose. (**A**), the modified strains of *agrobacteria* were grown on 1% agar with 0.05% yeast extract and 1% tryptone containing 0.005% aniline blue with varying levels of AE or glucose as indicated. (**B**), bacteria were grown in ADB broth (4% glucose) Cultures (50 mL) were shaken (150 rpm) in 500 mL flasks at 30 °C. Curdlan was harvested after 3 days (see materials and methods for details). Error bars indicate variation as standard deviation from three biological replicates. C58 did not produce any detectable curdlan.

## Data Availability

Data is contained within the article or Appendix A.

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
