# Peer review of "Genetic Engineering of *Agrobacterium* Increases Curdlan Production through Increased Expression of the *crdASC* Genes"

_microorganisms, 2023, doi:10.3390/microorganisms12010055_

Round 1
Reviewer 1 Report
Comments and Suggestions for Authors
The problem of obtaining highly productive strains using genetic and metabolic engineering methods is important and relevant. The article contains original ideas and approaches.
Some comments are:
1. In Introduction section many references are obsolete, please discuss your results with more recent papers.
2. Some taxa names are not in italicized. Italics are used for bacterial taxa at the level of family and below. Please correct here and elsewhere in the text.
Pg 2, line 76: Agrobacterium strains
Pg 3, line 90: Agrobacterium species
Pg 3, line 111: Agrobacterium sp. strain ATCC31749
3. Pg 2, line 51: «cellulose, which is produced by some species recoverable from soil, e.g., Gluconacetobacter xylinus……». Currently Gluconacetobacter xylinus belongs to the genus Komagataeibacter. Many strains of this genus were isolated from the kombucha community, vinegar, fruit, fruit juices, but not from the soil.
3. The aim of the work should be better justified according to the title of the article
«Genetic engineering of Agrobacterium increases curdlan production through increased expression of the crdASC genes».
«As one of the goals to this study, I wanted to investigate curdlan production by Agrobacterium species, in particular to see whether curdlan was also deposited as a discrete capsule on this bacterium. Reducing the cost of production was another goal of this study».
4. It would be beneficial to include information about the cost of the production process of curdlan in the text.
5. In Material and Methods section: «Media and culture conditions» and «Media for curdlan production» could be combined.
6. There is no statistical analysis in the Materials and Methods section.
7. Figure 6 A is not very clear.
8. The results of genetic modifications should be presented in the Results section.
9. Start the discussion with a paragraph that contains the main findings of the paper.
10. The conclusion should be highlighted in a separate part.
Author Response
- In Introduction section many references are obsolete, please discuss your results with more recent papers.
Response: The introduction now contains additional references, including some to very recent publications. See lines 61-64, 138-141, 182-184, 189-191.
- Some taxa names are not in italicized. Italics are used for bacterial taxa at the level of family and below. Please correct here and elsewhere in the text.
Response: The name Agrobacterium has been italicized throughout the manuscript.
- Pg 2, line 51: «cellulose, which is produced by some species recoverable from soil, e.g., Gluconacetobacter xylinus……». Currently Gluconacetobacter xylinus belongs to the genus Komagataeibacter. Many strains of this genus were isolated from the kombucha community, vinegar, fruit, fruit juices, but not from the soil.
Response: Phrase stating that Gluconeacetobacter xylinus is recoverable from soil has been removed.
- The aim of the work should be better justified according to the title of the article
«Genetic engineering of Agrobacterium increases curdlan production through increased expression of the crdASC genes».
«As one of the goals to this study, I wanted to investigate curdlan production by Agrobacterium species, in particular to see whether curdlan was also deposited as a discrete capsule on this bacterium. Reducing the cost of production was another goal of this study».
Response: The aim of the work was modified to fit better with the title of the manuscript. The text has been changed at Line 109 ‘Genetic engineering of Agrobacterium strains to increase curdlan production was another goal of this study.’
(4) . It would be beneficial to include information about the cost of the production process of curdlan in the text.
Response: I agree that this information would be interesting and useful. In fact, this is one of the projects that I am currently working on, with the specific goal of estimating the cost of curdlan production with and without genetically engineered strains. However, because the answer is not so simple, rather than adding this information here, I think it would be better to focus on this aspect in a follow-up manuscript. For example, in line 727, I have raised the prospect of varying curdlan yields after altering the fermentation parameters. These all impact the cost of production.
- In Material and Methods section: «Media and culture conditions» and «Media for curdlan production» could be combined.
Response: Done
- There is no statistical analysis in the Materials and Methods section.
Response: In this study, there was no official statistical analysis performed. All the experiments were performed at least three times, and standard deviation is routinely presented from at least 3 replicates (up to 8 replicates for the 96-well plate cultures).
- Figure 6 A is not very clear.
Response: The colour intensity is not meant to be quantitative but rather a qualitative indication of curdlan production (as stated in line 707). The photograph was made with a high quality camera. It cannot be improved. If recommended, I can remove the photo from the manuscript. However, I feel that the photo is valuable here because it provides an approximate indication of curdlan production that is easy to detect.
- The results of genetic modifications should be presented in the Results section.
Response: the outcome of the genetic modifications on curdlan production were included at the end of the results section (lines 582-589).
- Start the discussion with a paragraph that contains the main findings of the paper.
Response: the discussion now starts with a paragraph that contains the main findings of the work. (Line 591)
- The conclusion should be highlighted in a separate part.
Response: I have now added a new section 5. Conclusions and outlook at the end of the discussion. (Line 731)

Reviewer 2 Report
Comments and Suggestions for Authors
Q1: Here, EPS is an integral part of the biopolymer production,
Which genetic regulatory mechanisms are responsible for EPS production?
Please refer to: https://doi.org/10.1016/j.ecoenv.2023.115389
Lines 59-62: “……… curdlan production in the laboratory requires N-limited culture medium with ample carbohydrate, which fits with the general observation that the production of EPS by bacteria is frequently induced by nutritional stress e.g. N, P or S limitation [18-22].”
Q2: Since PHA production is also induced by by nutritional stress e.g. N, P or S limitation (Ref. https://doi.org/10.3390/polym15081937), Is there a co-relation between the Curdlan and polyhydroxyalkanote (PHA) biosynthetic pathways?
Lines 119-120: “… The crdASC genes form a gene cluster (i.e., all transcribed in the same 120 direction) ”
Are there any reports where the orientation of this operon is modified, and does it influence the curdlan production?
(Ref. to: https://doi.org/10.1016/j.biotechadv.2013.08.007).
Lines 181-183: “… a strong promoter named promoter PphaP 182 (promoter of gene phaP) from the Alphaproteobacterium Rhodobacter sphaeroides 183 [50, 51]..”
Is this promoter the same as used for polyhydroxyalkanote (PHA) biosynthesis?
Author Response
Q1: Here, EPS is an integral part of the biopolymer production,
Which genetic regulatory mechanisms are responsible for EPS production?
Please refer to: https://doi.org/10.1016/j.ecoenv.2023.115389
Response: The text has been modified (line 138) to point out that curdlan production differs from the general expectation that EPS production and biofilm formation in most bacteria are typically regulated by three primary mechanisms: quorum sensing, cyclic dinucleotides and small non-coding RNAs. Rather, upon encountering the environmental triggers for curdlan production (e.g., low nitrogen and abundant sugars) the transcription of the crdASC genes increases by almost 100-fold, and this depended upon phosphate accumulation and the stringent response (ppGpp) [38]. A new reference has been added.
Q2: Since PHA production is also induced by by nutritional stress e.g. N, P or S limitation (Ref. https://doi.org/10.3390/polym15081937), Is there a co-relation between the Curdlan and polyhydroxyalkanote (PHA) biosynthetic pathways?
Response: At line 61, the text now mentions that PHAs accumulate in other bacteria under the same conditions as curdlan production. However, to my knowledge, there has been no investigation into whether the industrial curdlan producer Agrobacterium sp. 31749 also produces PHA. It does contain the genes.
Q3: Are there any reports where the orientation of this operon is modified, and does it influence the curdlan production? (Ref. to: https://doi.org/10.1016/j.biotechadv.2013.08.007).
Response: There have been no reports in the literature where the region of the linear chromosome of Agrobacterium carrying the crdASC genes has been modified with the goal of increasing their expression. This statement has now been added to the text (line 182).
Q4: Lines 181-183: “… a strong promoter named promoter PphaP 182 (promoter of gene phaP) from the Alphaproteobacterium Rhodobacter sphaeroides 183 [50, 51]..”
Is this promoter the same as used for polyhydroxyalkanote (PHA) biosynthesis?
Response: The promoter PphaP is used to drive phasin production in Rhodobacter sphaeroides. Phasins are known to be important for PHA accumulation. This information has now been added to the text (line 190).

Round 2
Reviewer 1 Report
Comments and Suggestions for Authors
The author responded to all comments quite fully and changed the text of the article in accordance with the recommendations. But there are a few comments left.
1. I found only one new added reference at number 39 (2023) (Line 141).
Lines 61-64 contain obsolete references for the following years: 2000, 1983, 1990, 1988, 1998, 1990.
2. Lines 109, 595 – The name Agrobacterium was not italicized.
Author Response
- I found only one new added reference at number 39 (2023) (Line 141). Lines 61-64 contain obsolete references for the following years: 2000, 1983, 1990, 1988, 1998, 1990.
Response: Four newer references have now been added at line 61. Four of the older references have been removed.
2. Lines 109, 595 – The name Agrobacterium was not italicized.
Response: These instances of Agrobacterium have now been italicized.